# Learning Neural Representations of Human Cognition across Many fMRI Studies

**Arthur Mensch**[*]
Inria
`arthur.mensch@m4x.org`

**Julien Mairal**[†]
Inria
`julien.mairal@inria.fr`

**Danilo Bzdok**
Department of Psychiatry, RWTH
`danilo.bzdok@rwth-aachen.de`

**Bertrand Thirion**[*]
Inria
`bertrand.thirion@inria.fr`

**Gaël Varoquaux**[*]
Inria
`gael.varoquaux@inria.fr`

## Abstract

Cognitive neuroscience is enjoying rapid increase in extensive public brain-imaging datasets. It opens the door to large-scale statistical models. Finding a unified perspective for all available data calls for scalable and automated solutions to an old challenge: how to aggregate heterogeneous information on brain function into a universal cognitive system that relates mental operations/cognitive processes/psychological tasks to brain networks? We cast this challenge in a machine-learning approach to predict conditions from statistical brain maps across different studies. For this, we leverage multi-task learning and multi-scale dimension reduction to learn low-dimensional representations of brain images that carry cognitive information and can be robustly associated with psychological stimuli. Our multi-dataset classification model achieves the best prediction performance on several large reference datasets, compared to models without cognitive-aware low-dimension representations; it brings a substantial performance boost to the analysis of small datasets, and can be introspected to identify universal template cognitive concepts.

Due to the advent of functional brain-imaging technologies, cognitive neuroscience is accumulating quantitative maps of neural activity responses to specific tasks or stimuli. A rapidly increasing number of neuroimaging studies are publicly shared (*e.g.,* the human connectome project, HCP [1]), opening the door to applying large-scale statistical approaches [2]. Yet, it remains a major challenge to formally extract structured knowledge from heterogeneous neuroscience repositories. As stressed in [3], aggregating knowledge across cognitive neuroscience experiments is intrinsically difficult due to the diverse nature of the hypotheses and conclusions of the investigators. Cognitive neuroscience experiments aim at isolating brain effects underlying specific psychological processes: they yield statistical maps of brain activity that measure the neural responses to carefully designed stimulus. Unfortunately, neither regional brain responses nor experimental stimuli can be considered to be *atomic*: a given experimental stimulus recruits a spatially distributed set of brain regions [4], while each brain region is observed to react to diverse stimuli. Taking advantage of the resulting data richness to build formal models describing psychological processes requires to describe each cognitive

---

[*]Inria, CEA, Université Paris-Saclay, 91191 Gif sur Yvette, France
[†]Univ. Grenoble Alpes, Inria, CNRS, Grenoble INP, LJK, 38000 Grenoble, France

conclusion on a common basis for brain response and experimental study design. Uncovering *atomic basis functions* that capture the neural building blocks underlying cognitive processes is therefore a primary goal of neuroscience [5], for which we propose a new data-driven approach.

Several statistical approaches have been proposed to tackle the problem of knowledge aggregation in functional imaging. A first set of approaches relies on coordinate-based meta-analysis to define robust neural correlates of cognitive processes: those are extracted from the descriptions of experiments — based on categories defined by text mining [6] or expert [7]— and correlated with brain coordinates related to these experiments. Although quantitative meta-analysis techniques provide useful summaries of the existing literature, they are hindered by label noise in the experiment descriptions, and weak information on brain activation as the maps are reduced to a few coordinates [8]. A second, more recent set of approaches models directly brain maps across studies, either focusing on studies on similar cognitive processes [9], or tackling the entire scope of cognition [10, 11]. Decoding, *i.e.* predicting the cognitive process from brain activity, across many different studies touching different cognitive questions is a key goal for cognitive neuroimaging as it provides a principled answer to reverse inference [12]. However, a major roadblock to scaling this approach is the necessity to label cognitive tasks across studies in a rich but consistent way, *e.g.,* building an ontology [13].

We follow a more automated approach and cast dataset accumulation into a *multi-task learning problem*: our model is trained to decode simultaneously different datasets, using a shared architecture. Machine-learning techniques can indeed learn universal representations of inputs that give good performance in multiple supervised problems [14, 15]. They have been successful, especially with the development of deep neural network [see, *e.g.,* 16], in *sharing representations* and *transferring knowledge* from one dataset prediction model to another (*e.g.,* in computer-vision [17] and audio-processing [18]). A popular approach is to simultaneously learn to represent the inputs of the different datasets in a low-dimensional space and to predict the outputs from the low-dimensional representatives. Using very deep model architectures in functional MRI is currently thwarted by the signal-to-noise ratio of the available recordings and the relative little size of datasets [19] compared to computer vision and text corpora. Yet, we show that multi-dataset representation learning is a fertile ground for identifying cognitive systems with predictive power for mental operations.

**Contribution.** We introduce a new model architecture dedicated to multi-dataset classification, that performs two successive linear dimension reductions of the input statistical brain images and predicts psychological conditions from a *learned* low-dimensional representation of these images, linked to cognitive processes. In contrast to previous ontology-based approaches, imposing a structure across different cognitive experiments is not needed in our model: the representation of brain images is learned using the raw set of experimental conditions for each dataset. To our knowledge, this work is the first to propose knowledge aggregation and transfer learning in between functional MRI studies with such modest level of supervision. We demonstrate the performance of our model on several openly accessible and rich reference datasets in the brain-imaging domain. The different aspects of its architecture bring a substantial increase in out-of-sample accuracy compared to models that forgo learning a cognitive-aware low-dimensional representation of brain maps. Our model remains simple enough to be interpretable: it can be collapsed into a collection of classification maps, while the space of low-dimensional representatives can be explored to uncover a set of meaningful latent components.

## 1   Model: multi-dataset classification of brain statistical images

Our general goal is to extract and integrate biological knowledge across many brain-imaging studies within the same statistical learning framework. We first outline how analyzing large repositories of fMRI experiments can be cast as a classification problem. Here, success in capturing brain-behavior relationships is measured by out-of-sample prediction accuracy. The proposed model (Figure 1) solves a range of these classification problems in an identical statistical estimation and imposes a shared latent structure across the single-dataset classification parameters. These shared model parameters may be viewed as a chain of two dimension reductions. The first reduction layer leverages knowledge about brain spatial regularities; it is learned from resting-state data and designed to capture neural activity patterns at different coarseness levels. The second reduction layer projects data on directions generally relevant for cognitive-state prediction. The combination of both reductions yields low-dimensional representatives that are less affected by noise and subject variance than

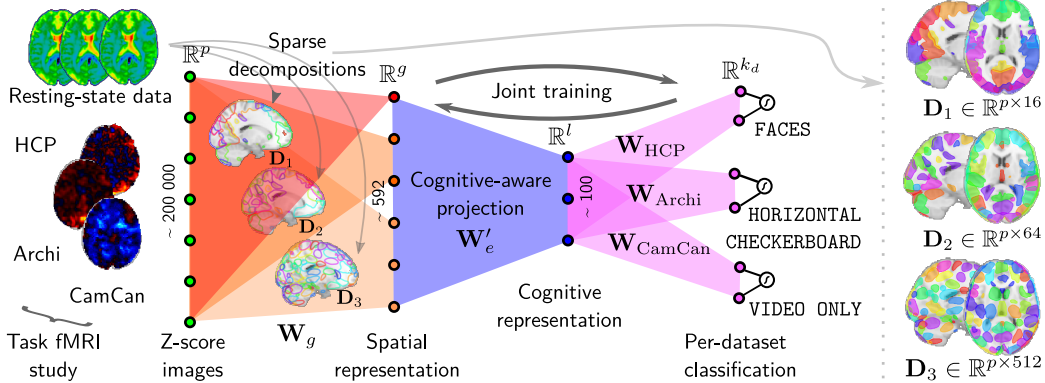

Figure 1: **Model architecture: Three-layer multi-dataset classification.** The first layer (*orange*) is learned from data acquired outside of cognitive experiments and captures a spatially coherent signal *at multiple scales*, the second layer (*blue*) embeds these representations in a space common to all datasets, from which the conditions are predicted (*pink*) from multinomial models.

the high-dimensional samples: classification is expected to have better out-of-sample prediction performance.

## 1.1   Problem setting: predicting conditions from brain activity in multiple studies

We first introduce our notations and terminology, and formalize a general prediction problem applicable to any task fMRI dataset. In a single fMRI *study*, each subject performs different *experiments* in the scanner. During such an experiment, the subjects are presented a set of sensory stimuli (*i.e.*, *conditions*) that aim at recruiting a target set of cognitive processes. We fit a first-level general linear model for every record to obtain z-score maps that quantify the importance of each condition in explaining each voxel. Formally, the $n$ statistical maps $(\mathbf{x}_i)_{i \in [n]}$ of a given study form a sequence in $\mathbb{R}^p$, where $p$ is the number of voxels in the brain. Each such observation is labelled by a condition $c_i$ in $[1, k]$ whose effect captures $\mathbf{x}_i$. A single study typically features one or a few (if experiments are repeated) statistical map per condition and per subject, and may present up to $k = 30$ conditions. Across the studies, the observed brain maps can be modeled as generated from an unknown joint distribution of brain activity and associated cognitive conditions $((\mathbf{x}_i, c_i))_{i \in [n]}$ where variability across trials and subjects acts as confounding noise. In this context, we wish to learn a decoding model that predicts condition $c$ from brain activity $\mathbf{x}$ measured from new subjects or new studies.

Inspired by recent work [10, 20, 21], we frame the condition prediction problem into the estimation of a multinomial classification model. Our models estimate a probability vector of $\mathbf{x}$ being labeled by each condition in $C$. This vector is modeled as a function of $(\mathbf{W}, \mathbf{b})$ in $\mathbb{R}^{p \times k} \times \mathbb{R}^k$ that takes the *softmax* form. For all $j$ in $[1, k]$, its $j$-th coordinate is defined as

$$\mathbf{p}(\mathbf{x}, \mathbf{W}, \mathbf{b})_j \triangleq \mathbb{P}[c = j | \mathbf{x}, \mathbf{W}, \mathbf{b}] = \frac{e^{\mathbf{W}^{(j)\top}\mathbf{x} + \mathbf{b}}}{\sum_{l \in C} e^{\mathbf{W}^{(l)\top}\mathbf{x} + \mathbf{b}}}. \tag{1}$$

Fitting the model weights is done by minimizing the cross-entropy between $(\mathbf{p}(\mathbf{x}_i))_i$ and the true labels $([c_i = j]_{j \in [k]})_i$, with respect to $(\mathbf{W}, \mathbf{b})$, with or without imposing parameter regularization. In this model, an input image is classified in between all conditions presented in the whole *study*. It is possible to restrict this classification to the set of conditions used in a given *experiment* — the empirical results of this study can be reproduced in this setting.

**The challenge of model parameter estimation.**   A major inconvenience of the vanilla multinomial model lies in the ratio between the limited number of samples provided by a typical fMRI dataset and the overwhelming number of model weights to be estimated. Fitting the model amounts to estimating $k$ discriminative brain map, *i.e.* millions of parameters (4M for the 23 conditions of HCP), whereas most brain-imaging studies yield less than a hundred observations and therefore only a few thousands samples. This makes it hard to reasonably approximate the population parameters for successful generalization, especially because the variance between subjects is high compared to the

variance between conditions. The obstacle is usually tackled in one of two major ways in brain-imaging: 1) we can impose sparsity or a-priori structure over the model weights. Alternatively, 2) we can reduce the dimension of input data by performing spatial clustering or univariate feature selection by ANOVA. However, we note that, on the one hand, regularization strategies frequently incur costly computational budgets if one wants to obtain interpretable weights [22] and they introduce artificial bias. On the other hand, existing techniques developed in fMRI analysis for dimension reduction can lead to distorted signal and accuracy losses [23]. Most importantly, previous statistical approaches are not tuned to identifying conditions from task fMRI data. We therefore propose to use a dimension reduction that is *estimated* from data and tuned to capture the common hidden aspects shared by statistical maps across studies — we aggregate several classification models that share parameters.

## 1.2 Learning shared representation across studies for decoding

We now consider several fMRI studies. $(\mathbf{x}_i)_{i \in [n]}$ is the union of all statistical maps from all datasets. We write $D$ the set of all studies, $C_d$ the set of all $k_d$ conditions from study $d$, $k \triangleq \sum_d k_d$ the total number of conditions and $S_d$ the subset of $[n]$ that index samples of study $d$. For each study $d$, we estimate the parameters $(\mathbf{W}_d, \mathbf{b}_d)$ for the classification problem defined above. Adapting the multi-task learning framework of [14], we constrain the weights $(\mathbf{W}_d)_d$ to share a common latent structure: namely, we fix a latent dimension $l \leq p$, and enforce that for all datasets $d$,

$$\mathbf{W}_d = \mathbf{W}_e \mathbf{W}'_d, \tag{2}$$

where the matrix $\mathbf{W}_e$ in $\mathbb{R}^{p \times l}$ is shared across datasets, and $(\mathbf{W}'_d)_d$ are dataset-specific classification matrices from a $l$ dimensional input space. Intuitively, $\mathbf{W}_e$ should be a "consensus" projection matrix, that project every sample $\mathbf{x}_i$ from every dataset onto a lower dimensional representation $\mathbf{W}_e^\top \mathbf{x}_i$ in $\mathbb{R}^l$ that is easy to label correctly.

The latent dimension $l$ may be chosen larger than $k$. In this case, regularization is necessary to ensure that the factorization (2) is indeed useful, *i.e.*, that the multi-dataset classification problem does not reduce to separate multinomial regressions on each dataset. To regularize our model, we apply *Dropout* [24] to the projected data representation. Namely, during successive training iterations, we set a random fraction $r$ of the reduced data features to 0. This prevents the co-adaptation of matrices $\mathbf{W}_e$ and $(\mathbf{W}'_d)_d$ and ensures that every direction of $\mathbf{W}_e$ is useful for classifying every dataset. Formally, Dropout amounts to sample binary diagonal matrices $\mathbf{M}$ in $\mathbb{R}^{l \times l}$ during training, with Bernouilli distributed coefficients; for all datasets $d$, $\mathbf{W}'_d$ is estimated through the task of classifying Dropout-corrupted reduced data $(\mathbf{M}\mathbf{W}_e^\top \mathbf{x}_i)_{i \in S_d, \mathbf{M} \sim \mathcal{M}}$.

In practice, matrices $\mathbf{W}_e$ and $(\mathbf{W}'_d)_d$ are learned by jointly minimizing the following expected risk, where the objective is the sum of each of single-study cross-entropies, averaged over Dropout noise:

$$\min_{\substack{\mathbf{W}_e \\ (\mathbf{W}'_d)_d}} \sum_{d \in D} \frac{1}{|S_d|} \sum_{i \in S_d} \sum_{j \in C_d} \mathbb{E}_\mathbf{M} \big[ -\delta_{j=c_i} \log \mathbf{p}_d[\mathbf{x}_i, \mathbf{W}_e \mathbf{M} \mathbf{W}'_d, \mathbf{b}_d]_j] \big] \tag{3}$$

Imposing a common structure to the classification matrices $(\mathbf{W}_d)_d$ is natural as the classes to be distinguished do share some common neural organization — brain maps have a correlated spatial structure, while the psychological conditions of the diffent datasets may trigger shared cognitive primitives underlying human cognition [21, 20]. With our design, we aim at learning a matrix $\mathbf{W}_e$ that captures these common aspects and thus benefits the generalization performance of *all* the classifiers. As $\mathbf{W}_e$ is *estimated* from data, brain maps from one study are enriched by the maps from all the other studies, even if the conditions to be classified are not shared among studies. In so doing, our modeling approach allows *transfer learning* among all the classification tasks.

Unfortunately, estimators provided by solving (3) may have limited generalization performance as $n$ remain relatively small ($\sim 20,000$) compared to the number of parameters. We address this issue by performing an initial dimension reduction that captures the spatial structure of brain maps.

## 1.3 Initial dimension reduction using localized rest-fMRI activity patterns

The projection expressed by $\mathbf{W}_e$ ignores the signal structure of statistical brain maps. Acknowledging this structure in commonly acquired brain measurements should allow to reduce the dimensionality of data with little signal loss, and possibly the additional benefit of a denoising effect. Several recent

studies [25] in the brain-imaging domain suggest to use fMRI data acquired in experiment-free studies for such dimension reduction. For this reason, we introduce a first reduction of dimension that is not estimated from statistical maps, but from resting-state data. Formally, we enforce $\mathbf{W}_e = \mathbf{W}_g \mathbf{W}'_e$, where $g > l$ ($g \sim 300$), $\mathbf{W}_g \in \mathbb{R}^{p \times g}$ and $\mathbf{W}'_e \in \mathbb{R}^{g \times k}$. Intuitively, the multiplication by matrix $\mathbf{W}_g$ should summarize the spatial distribution of brain maps, while multiplying by $\mathbf{W}'_e$, that is estimated solving (3), should find low-dimensional representations able to capture cognitive features. $\mathbf{W}'_e$ is now of reasonable size ($g \times l \sim 15000$): solving (3) should estimate parameters with better generalization performance. Defining an appropriate matrix $\mathbf{W}_g$ is the purpose of the next paragaphs.

**Resting-state decomposition.** The initial dimension reduction determines the relative contribution of statistical brain maps over what is commonly interpreted by neuroscience investigators as *functional networks*. We discover such macroscopical brain networks by performing a sparse matrix factorization over the massive resting-state dataset provided in the HCP900 release [1]: such a decomposition technique, described *e.g.,* in [26, 27] efficiently provides (*i.e.*, in the order of few hours) a given number of sparse spatial maps that decompose the resting state signal with good reconstruction performance. That is, it finds a *sparse* and *positive* matrix $\mathbf{D}$ in $\mathbb{R}^{p \times g}$ and loadings $\mathbf{A}$ in $\mathbb{R}^{g \times m}$ such that the $m$ resting-state brain images $\mathbf{X}_{rs}$ in $\mathbb{R}^{p \times m}$ are well approximated by $\mathbf{DA}$. $\mathbf{D}$ is this a set of slightly overlapping networks — each voxel belongs to at most two networks. To maximally preserve Euclidian distance when performing the reduction, we perform an *orthogonal* projection, which amounts to setting $\mathbf{W}_g \triangleq \mathbf{D}(\mathbf{D}^\top \mathbf{D})^{-1}$. Replacing in (3), we obtain the reduced expected risk minimization problem, where the input dimension is now the number $g$ of dictionary components:

$$\min_{\substack{\mathbf{W}'_e \in \mathbb{R}^{g \times l} \\ (\mathbf{W}'_d)_d}} \sum_{d \in D} \frac{1}{|S_d|} \sum_{i \in S_d} \sum_{j \in C_d} \mathbb{E}_{\mathbf{M}} \big[ - \delta_{j=c_i} \log \mathbf{p}_d [\mathbf{W}_g^\top \mathbf{x}_i, \mathbf{W}'_e \mathbf{M} \mathbf{W}'_d, \mathbf{b}_d]_j \big]. \qquad (4)$$

**Multiscale projection.** Selecting the "best" number of brain networks $q$ is an ill-posed problem [28]: the size of functional networks that will prove relevant for condition classification is unknown to the investigator. To address this issue, we propose to reduce high-resolution data $(\mathbf{x}_i)_i$ in a multi-scale fashion: we initially extract 3 sparse spatial *dictionaries* $(\mathbf{D}_j)_{j \in [3]}$ with 16, 64 and 512 components respectively. Then, we project statistical maps onto each of the dictionaries, and concatenate the loadings, in a process analogous to projecting on an overcomplete dictionary in computer vision [*e.g.,* 29]. This amounts to define the matrix $\mathbf{W}_g$ as the concatenation

$$\mathbf{W}_g \triangleq [\mathbf{D}_1(\mathbf{D}_1^\top \mathbf{D}_1)^{-1} \; \mathbf{D}_2(\mathbf{D}_2^\top \mathbf{D}_2)^{-1} \; \mathbf{D}_3(\mathbf{D}_3^\top \mathbf{D}_3)^{-1}] \in \mathbb{R}^{p \times (16+64+512)}. \qquad (5)$$

With this definition, the reduced data $(\mathbf{W}_g^\top \mathbf{x}_i)_i$ carry information about the network activations at different scales. As such, it makes the classification maps learned by the model more regular than when using a single-scale dictionary, and indeed yields more interpretable classification maps. However, it only brings only a small improvement in term of predictive accuracy, compared to using a simple dictionary of size $k = 512$. We further discuss multi-scale decomposition in Appendix A.2.

### 1.4 Training with stochastic gradient descent

As illustrated in Figure 1, our model may be interpreted as a three-layer neural network with linear activations and several read-out heads, each corresponding to a specific dataset. The model can be trained using stochastic gradient descent, following a previously employed alternated training scheme [18]: we cycle through datasets $d \in D$ and select, at each iteration, a mini-batch of samples $(\mathbf{x}_i)_{i \in B}$, where $B \subset S_d$ has the same size for all datasets. We perform a gradient step — the weights $\mathbf{W}'_d, \mathbf{b}_d$ and $\mathbf{W}'_e$ are updated, while the others are left unchanged. The optimizer thus sees the same number of samples for each dataset, and the expected stochastic gradient is the gradient of (4), so that the empirical risk decreases in expectation and we find a critical point of (4) asymptotically. We use the Adam solver [30] as a flavor of stochastic gradient descent, as it allows faster convergence.

**Computational cost.** Training the model on projected data $(\mathbf{W}_g^\top \mathbf{x}_i)_i$ takes 10 minutes on a conventional single CPU machine with an Intel Xeon 3.21Ghz. The initial step of computing the dictionaries $(\mathbf{D}_1, \mathbf{D}_2, \mathbf{D}_3)$ from all HCP900 resting-state (4TB of data) records takes 5 hours using [27], while transforming data from all the studies with $\mathbf{W}_g$ projection takes around 1 hour. Adding a new dataset with 30 subjects to our model and performing the joint training takes no more than 20 minutes. This is much less than the cost of fitting a first-level GLM on this dataset ($\sim$ 1h per subject).

## 2   Experiments

We characterize the behavior and performance of our model on several large, publicly available brain-imaging datasets. First, to validate the relevance of all the elements of our model, we perform an ablation study. It proves that the multi-scale spatial dimension reduction and the use of multi-dataset classification improves substancially classification performance, and suggests that the proposed model captures a new interesting latent structure of brain images. We further illustrate the effect of *transfer learning*, by systematically varying the number of subjects in a single dataset: we show how multi-dataset learning helps mitigating the decrease in accuracy due to smaller train size — a result of much use for analysing cognitive experiments on small cohorts. Finally, we illustrate the interpretability of our model and show how the latent "cognitive-space" can be explored to uncover some template brain maps associated with related conditions in different datasets.

### 2.1   Datasets and tools

**Datasets.**   Our experimental study features 5 publicly-available task fMRI study. We use all resting-state records from the HCP900 release [1] to compute the sparse dictionaries that are used in the first dimension reduction materialized by $\mathbf{W}_g$. We succinctly describe the conditions of each dataset — we refer the reader to the original publications for further details.

- **HCP**: gambling, working memory, motor, language, social and relational tasks. 800 subjects.
- **Archi** [31]: localizer protocol, motor, social and relational task. 79 subjects.
- **Brainomics** [32]: localizer protocol. 98 subjects.
- **Camcan** [33]: audio-video task, with frequency variation. 606 subjects.
- **LA5c consortium** [34]: task-switching, balloon analog risk taking, stop-signal and spatial working memory capacity tasks — high-level tasks. 200 subjects.

The last four datasets are *target datasets*, on which we measure out-of-sample prediction performance. The larger HCP dataset serves as a *knowledge transfering dataset*, which should boost these performance when considered in the multi-dataset model. We register the task time-series in the reference MNI space before fitting a general linear model (GLM) and computing the maps (standardized by z-scoring) associated with each *base* condition — no manual design of contrast is involved. More details on the pipeline used for z-map extraction is provided in Appendix A.1.

**Tools.**   We use pytorch [1] to define and train the proposed models, nilearn [35] to handle brain datasets, along with scikit-learn [36] to design the experimental pipelines. Sparse brain decompositions were computed from the whole HCP900 resting-state data. The code for reproducing experiments is available at `http://github.com/arthurmensch/cogspaces`. Our model involves a few non-critical hyperparameters: we use batches of size $256$, set the latent dimension $l = 100$ and use a Dropout rate $r = 0.75$ in the latent cognitive space — this value perform slightly better than $r = 0.5$. We use a multi-scale dictionary with $16$, $64$ and $512$ components, as it yields the best quantitative and qualitative results.[2] Finally, test accuracy is measured on half of the subjects of each dataset, that are removed from training sets beforehand. Benchmarks are repeated 20 times with random split folds to estimate the variance in performance.

### 2.2   Dimension reduction and transfer improves test accuracy

For the four benchmark studies, the proposed model brings between +1.3% to +13.4% extra test accuracy compared to a simple multinomial classification. To further quantify which aspects of the model improve performance, we perform an ablation study: we measure the prediction accuracy of six models, from the simplest to the most complete model described in Section 1. The first three experiments study the effect of initial dimension reduction and regularization[3]. The last three experiments measure the performance of the proposed factored model, and the effect of multi-dataset classification.

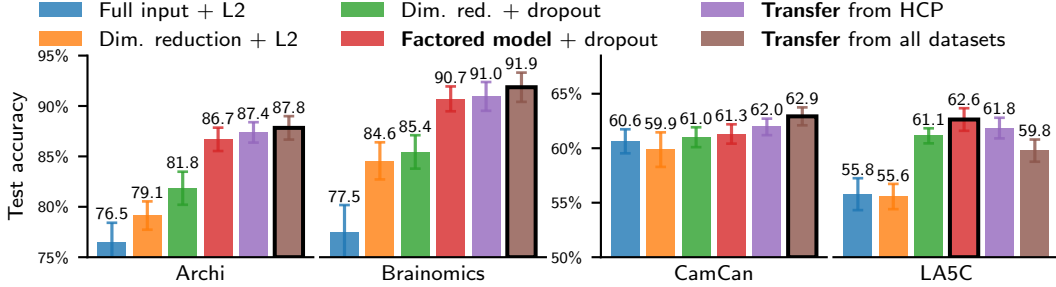

Figure 2: **Ablation results.** Each dimension reduction of the model has a relevant contribution. Dropout regularization is very effective when applied to the cognitive latent space. Learning this latent space allows to transfer knowledge between datasets.

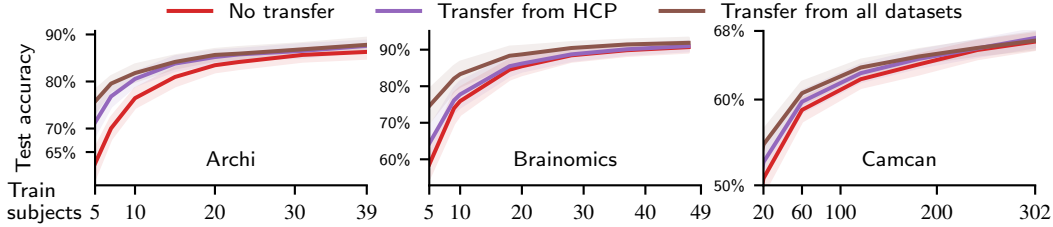

Figure 3: **Learning curves** in the single-dataset and multi-dataset setting. Estimating the latent cognitive space from multiple datasets is very useful for studying small cohorts.

1. Baseline $\ell_2$-penalized multinomial classification, where we predict $c$ from $\mathbf{x} \in \mathbb{R}^p$ directly.
2. Multinomial classification after projection on a dictionary, *i.e.* predicting $c$ from $\mathbf{W}_g\mathbf{x}$.
3. Same as experience 2, using Dropout noise on projected data $\mathbf{W}_g\mathbf{x}$.
4. Factored model in the single-study case: solving (4) with the target study only.
5. Factored model in a two-study case: using target study alongside HCP.
6. Factored model in the *multi*-study case: using target study alongside all other studies.

The results are summarized in Figure 2. On average, both dimension reduction introduced by $\mathbf{W}_g$ and $\mathbf{W}'_e$ are beneficial to generalization performance. Using many datasets for prediction brings a further increase in performance, providing evidence of transfer learning between datasets.

In detail, the comparison between experiments 1, 2 and 3 confirms that projecting brain images onto functional networks of interest is a good strategy to capture cognitive information [20, 25]. Note that in addition to improving the statistical properties of the estimators, the projection reduces drastically the computational complexity of training our full model. Experiment 2 and 3 measure the impact of the regularization method *without* learning a further latent projection. Using Dropout on the input space performs consistently better than $\ell_2$ regularization ($+\mathbf{1}\%$ to $+\mathbf{5}\%$); this can be explained in view of [37], that interpret input-Dropout as a $\ell_2$ regularization on the natural model parametrization.

Experiment 4 shows that Dropout regularization becomes much more powerful when learning a second dimension reduction, *i.e.* when solving problem (4). Even when using a single study for learning, we observe a significant improvement ($+\mathbf{3}\%$ to $+\mathbf{7}\%$) in performance on three out of four datasets. Learning a latent space projection together with Dropout-based data augmentation in this space is thus a much better regularization strategy than a simple $\ell_2$ or input-Dropout regularization.

Finally, the comparison between experiments 4, 5 and 6 exhibits the expected *transfer* effect. On three out of four target studies, learning the projection matrix $\mathbf{W}'_e$ using several datasets leads to an accuracy gain from $+\mathbf{1.1}\%$ to $+\mathbf{1.6}\%$, consistent across folds. The more datasets are used, the higher the accuracy gain — already note that this gain increases with smaller train size. Jointly classifying images on several datasets thus brings extra information to the cognitive model, which allows to find better representative brain maps for the target study. In particular, we conjecture that the large number of subjects in HCP helps modeling inter-subject noises. On the other hand, we observe a *negative* transfer effect on LA5c, as the tasks of this dataset share little cognitive aspects with the tasks of the other datasets. This encourages us to use richer dataset repositories for further improvement.

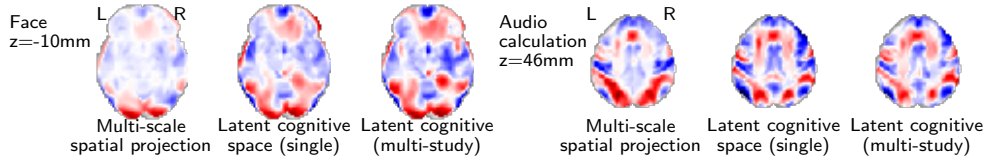

Figure 4: Classification maps from our model are more specific of higher level functions: they focus more on the FFA for faces, and on the *left* intraparietal suci for calculations.

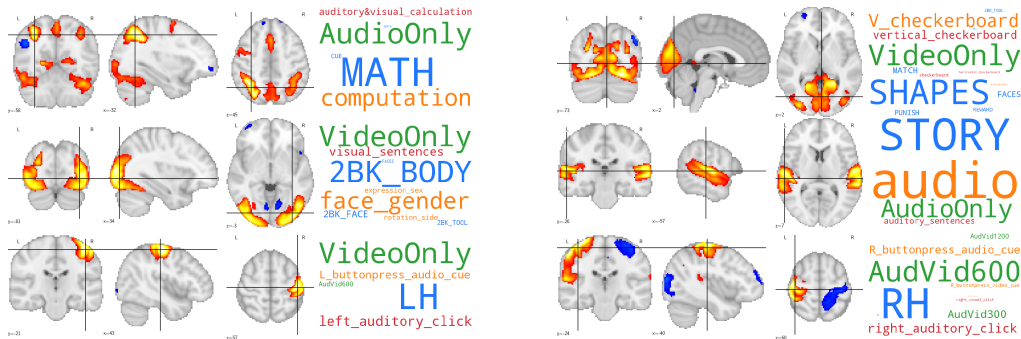

Figure 5: The latent space of our model can be explored to unveil some template brain statistical maps, that corresponds to bags of conditions related across color-coded datasets.

## 2.3 Transfer learning is very effective on small datasets

To further demonstrate the benefits of the multi-dataset model, we vary the size of target datasets (Archi, Brainomics and CamCan) and compare the performance of the single-study model with the model that aggregates Archi, Brainomics, CamCan and HCP studies. Figure 3 shows that the effect of transfer learning increases as we reduce the training size of the target dataset. This suggests that the learned data embedding $\mathbf{W}_g \mathbf{W}'_e$ does capture some universal cognitive information, and can be learned from different data sources. As a consequence, aggregating a larger study to mitigate the small number of training samples in the target dataset. With only 5 subjects, the gain in accuracy due to transfer is $+13\%$ on Archi, $+8\%$ on Brainomics, and $+6\%$ on CamCan. Multi-study learning should thus proves very useful to classify conditions in studies with ten or so subjects, which are still very common in neuroimaging.

## 2.4 Introspecting classification maps

At prediction time, our multi-dataset model can be collapsed into one multinomial model per dataset. Each dataset $d$ is then classified using matrix $\mathbf{W}_g \mathbf{W}'_e \mathbf{W}'_d$. Similar to the linear models classically used for decoding, the model weights for each condition can be represented as a brain map. Figure 4 shows the maps associated with digit computation and face viewing, for the Archi dataset. The models 2, 4 and 5 from the ablation study are compared. Although it is hard to assess the intrinsic quality of the maps, we can see that the introduction of the second projection layer and the multi-study problem formulation (here, appending the HCP dataset) yields maps with more weight on the high-level functional regions known to be specific of the task: for face viewing, the FFA stands out more compared to primary visual cortices; for calculations, the weights of the intraparietal sulci becomes left lateralized, as it has been reported for symbolic number processing [38].

## 2.5 Exploring the latent space

Within our model, classification is performed on the same $l$-dimensional space $E$ for all datasets, that is learned during training. To further show that this space captures some cognitive information, we extract from $E$ template brain images associated to general cognitive concepts. Fitting our model on the Archi, Brainomics, CamCan and HCP studies, we extract representative vectors of $E$ with a k-means clustering over the projected data and consider the centroids $(\mathbf{y}_j)_j$ of 50 clusters. Each centroid $\mathbf{y}_j$ can be associated to a brain image $\mathbf{t}_j \in \mathbb{R}^p$ that lies in the span of $\mathbf{D}_1, \mathbf{D}_2$

and $\mathbf{D}_3$. In doing so, we go backward through the model and obtain a representative of $\mathbf{y}_j$ with well delineated spatial regions. Going forward, we compute the classification probability vectors $\mathbf{W}_d^\top \mathbf{y}_j = {\mathbf{W}_d'}^\top {\mathbf{W}_e'}^\top \mathbf{W}_g^\top \mathbf{t}_j$ for each study $d$. Together, these probability vectors give an indication on the cognitive functions that $\mathbf{t}_j$ captures. Figure 5 represents six template images, associated to their probability vectors, shown as word clouds. We clearly obtain interpretable pairs of brain image/cognitive concepts. These pairs capture across datasets clusters of experiment conditions with similar brain representations.

# 3    Discussion

We compare our model to a previously proposed formulation for brain image classification. We show how our model differs from convex multi-task learning, and stress the importance of Dropout.

**Task fMRI classification.**   Our model is related to a previous semi-supervised classification model [20] that also performs multinomial classification of conditions in a low-dimensional space: the dimension reduction they propose is the equivalent of our projection $\mathbf{W}_g$. Our approach differs in two aspects. First, we replace the initial semi-supervised dimension reduction with unsupervised analysis of resting-state, using a much more tractable approach that we have shown to be conservative of cognitive signals. Second, we introduce the additional cognitive-aware projection $\mathbf{W}_e'$, learned on multiple studies. It substancially improves out-of-sample prediction performance, especially on small datasets, and above all allow to uncover a cognitive-aware latent space, as we have shown in our experiments.

**Convex multi-task learning.**   Due to the Dropout regularization and the fact that $l$ is allowed to be larger than $k$, our formulation differs from the classical approach [39] to the multi-task problem, that would estimate $\mathbf{\Theta} = \mathbf{W}_e'[\mathbf{W}_1', \ldots, \mathbf{W}_d']_d \in \mathbb{R}^{g \times k}$ by solving a convex empirical risk minimization problem with a trace-norm penalization, that encourages $\mathbf{\Theta}$ to be low-rank. We tested this formulation, which does not perform better than the explicit factorization formulation with Dropout regularization. Trace-norm regularized regression has the further drawback of being slower to train, as it typically operates with full gradients, *e.g.* using FISTA [40]. In contrast, the non-convex explicit factorization model is easily amenable to large-scale stochastic optimization — hence our focus.

**Importance of Dropout.**   The use of Dropout regularization is crucial in our model. *Without* Dropout, in the single-study case with $l > k$, solving the factored problem (4) yields a solution worse in term of empirical risk than solving the simple multinomial problem on $\left(\mathbf{W}_g^\top \mathbf{x}_i\right)_i$, which finds a global minimizer of (4). Yet, Figure 2 shows that the model enriched with a latent space (*red*) has better performance in test accuracy than the simple model (*orange*), thanks to the Dropout noise applied to the latent-space representation of the input data. Dropout is thus a promising novel way of regularizing fMRI models.

# 4    Conclusion

We proposed and characterized a novel cognitive neuroimaging modeling scheme that blends latent factor discovery and transfer learning. It can be applied to many different cognitive studies jointly without requiring explicit correspondences between the cognitive tasks. The model helps identifying the fundamental building blocks underlying the diversity of cognitive processes that the human mind can realize. It produces a basis of cognitive processes whose generalization power is validated quantitatively, and extracts representations of brain activity that grounds the transfer of knowledge from existing fMRI repositories to newly acquired task data. The captured cognitive representations will improve as we provide the model with a growing number of studies and cognitive conditions.

# 5    Acknowledgments

This project has received funding from the European Union's Horizon 2020 Framework Programme for Research and Innovation under grant agreement N° 720270 (Human Brain Project SGA1). Julien Mairal was supported by the ERC grant SOLARIS (N° 714381) and a grant from ANR (MACARON project ANR-14-CE23-0003-01). We thank Olivier Grisel for his most helpful insights.

## Footnotes

[1] `http://pytorch.org/`

[2] Note that using only the 512-components dictionary yields comparable predictive accuracy. Quantitatively, the multi-scale approach is beneficial when using dictionary with less components (*e.g.*, 16, 64, 128) — see Appendix A.2 for a quantitative validation of the multi-scale approach.

[3] For these models, $\ell_2$ and Dropout regularization parameter are estimated by nested cross-validation.

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
