[Supplementary Material]

# A  Supplementary material

## A.1  Data processing pipeline

**Resting-state fMRI.**   To extract sparse dictionaries, we perform dictionary learning on resting state time-series from the 900 subject of the HCP900 dataset. These time series are already provided preprocessed and aligned in the MNI space. We apply a smoothing kernel of size 4 mm to the data. We then use the modl[4] library to extract dictionaries from data by streaming records (in Nifti format) to an online algorithm. We run the algorithm on a grid of regularization parameters, that control the sparsity. We then select the sparsest dictionaries that cover all the brain. Dictionaries have been uploaded on our reproduction repository for potential further use.

**Task fMRI.**   To construct task fMRI datasets (HCP, Archi, Brainomics, Camcan, LA5C), we perform as follow. We use SPM (via the pypreprocess library[5]) for standard preprocessing of task fMRI time series: motion correction, slice-time interpolation and common registration to MNI space, using the DARTEL toolbox [1]. We use a smoothing kernel of size 4 mm. We then fit a first level general linear model to each subject. We use the result of the regression to obtain one z-statistic map per base condition (a.k.a. explatory variable) and per record. This pipeline can be streamlined for fMRI datasets provided in BIDS [2] format, using the nistats library[6].

Figure 6: Using a multi-scale dictionaries consistently provide better performance than single-scale dictionaries for classical dictionary sizes.

## A.2  Multi-scale dictionaries

To further investigate the interest of multi-scale projection on resting-state dictionaries, we enforce a more agressive geometric dimension reduction in our model than in the main text. That is, we use dictionaries with fewer components than in Section 2. This is typically a way to improve interpretability of the model by sacrifying a little out-of-sample accuracy. It is also adviseable to restrain the size of the dictionary if ones use a smaller resting-state dataset than HCP — large dictionaries tend to overfit small datasets.

In practice, we compare the dimension reduction using a 128 components dictionary to the dimension reduction using the same dictionary, along with a 16 and a 64 component dictionary. The results are presented in Figure 6. We observe that multi-scale projection performs systematically better than single scale projection. Note that the difference between the performance of the two sets of model is significant as it consistent across all folds. The multi-scale approach is thus quantitatively useful when using relatively small dictionaries from resting state data.

## Extra references

[1] John Ashburner. A fast diffeomorphic image registration algorithm. *Neuroimage*, 38(1):95–113, 2007.

[2] Krzysztof J. Gorgolewski, Tibor Auer, Vince D. Calhoun, R. Cameron Craddock, Samir Das, Eugene P. Duff, Guillaume Flandin, Satrajit S. Ghosh, Tristan Glatard, Yaroslav O. Halchenko, Daniel A. Handwerker, Michael Hanke, David Keator, Xiangrui Li, Zachary Michael, Camille Maumet, B. Nolan Nichols, Thomas E. Nichols, John Pellman, Jean-Baptiste Poline, Ariel Rokem, Gunnar Schaefer, Vanessa Sochat, William Triplett, Jessica A. Turner, Gaël Varoquaux, and Russell A. Poldrack. The brain imaging data structure, a format for organizing and describing outputs of neuroimaging experiments. *Scientific Data*, 3:160044, 2016.

## Footnotes

[4]http://github.com/arthurmensch/modl

[5]http://github.com/neurospin/pypreprocess

[6]http://github.com/nistats/nistats