[Reviews · NeurIPS 2017]

Reviewer 1



This paper proposes a new model architecture dedicated to multi-dataset brain decoding classification. Multi-dataset classification is a tricky problem in machine learning, especially when the number of samples is particularly small. In order to solve this problem, the author(s) employed the ideas of knowledge aggregation and transfer learning. The main idea of this paper is interesting but my main concerts are on the limited novelty compared to the previous work. Furthermore, I do not find any references or discussions in order to present the limitation of the proposed methods. Some reconstructive comments are listed as follows: 1. The main problem of this paper is the empirical studies. Author(s) must discuss in details how they registered and matched the different size of fMRI images to a common space in order to employ them for learning! Further, they must explain how different cognitive states are extracted from task-based fMRI datasets in order to combine them with rest-mode fMRI features! 2. Another problem in the empirical studies is that this paper did not compare the performance of the proposed method with state-of-the-arts methods applied to each of employed datasets! It could be interesting to figure out what new embedded information can be discovered by combining these datasets. 3. Another problem is the generality of the proposed methods. In order to apply the proposed method to other existed datasets, Author(s) must present more technical issues and general problems that could be happened for combining these datasets, e.g. the challenge of removing noise and sparsity, or even matching the same cognitive tasks from different datasets, etc. 4. In Figure 4, what is the maximum and minimum of correlation? It is better to show the fluctuation of values in this figure by using a bar (from dark red to dark blue). 5. As another problem in Figure 4, why all regions of the human brain are activated for a singular task? 6. The error bars must be reported in Figure 2 and Figure 3.

Reviewer 2



This paper introduces a framework to analyze heterogeneous fMRI datasets combining multi-task supervised learning with multi-scale dictionary learning. The result shows that classification of task conditions in a target dataset can be improved by integrating the large-scale database of humman connectome project (HCP) using their method. The proposed entire framework looks fairly reasonable to me. However, the method contains many different technical ideas and each of them is not necessarily well validated. The technical ideas include at least the following: 1) use of resting state data instead of task data for training the first stage; 2) use of multi-scale spatial dictionary combining multiple runs of sparse dictionary learning; 3) using Dropout for regularization; and 4) adding HCP data to improve classification on another target dataset. In my view, current results only support point 4. The following comments are related to the other points. In the experiment (Sec2.2), the comparison between single and multi-scale dictionaries (methods 2 and 3) seems to be made for different dictionary sizes (256 and 336). However, I think they should be balanced for a fair comarison (for instance, learn 336 atoms at once for the non-multi-scale dictionary), since currently the effects of having different dimensionalities and different spatial scales cannot be dissociated. I would also suggest comparing with simply using PCA to reduce to the same dimensionality, which may clarify the advantage of using localized features. Moreover, the methods 1-3 use L2-regularization but 4-5 uses Dropout, which may also not be a fair comparison. What happens if you use L2-regualarization for methods 4-5, or otherwise Dropout for 1-3? I also wondered what happens if the spatial dictionary is trained based on the task data rather than resting-state data. The use of rest data for training sounds reasonable for the goal of "uncovering atomic bases" (Introduction) but it would be necessary to compare the two cases in terms of both test accuracy and interpretability. In particular, can similar spatial maps in Figs4,5 be obtained even without resting-state decomposition? Other minor comments: - Eq1,3: what does the objective actually mean, when taking expectation with respect to M? Is it correct that the objective is minimized rather than maximized? - Eq3: "log" seems to be dropped. - Fig2: what are the chance levels for each dataset? No result by "full multinomial" for LA5c. - Line235: what does "base condition" mean?

Reviewer 3



This work presents a multi-task learning approach to meta-analysis in functional neuroimaging (fmri). The implementation and evaluation steps are very careful, evidence for benefits of transfer learning are presented. Results reveal that tricks from deep learning (stochastic optimization and stochastic regularization / drop-out) are useful - also for these medium scale, linear estimation problems. A main weakness is the starting point of using z-score maps as representation of the data. While it may be a productive reduction of complexity, it would have been nice to see a critical discussion of this step. A few other opportunities a missed: There is limited formal quantification of uncertainty (modeling/Bayes, resampling) There is no attempt of integrating the label structures across studies (i.e. borrowing power from similar labels wrt cognitive structure/ontology) There is limited attempt to place the work in a greater context of neuroimage meta-analysis. The literature review misses relevant work on meta-analysis e.g. based on coordinate based reconstructed activity maps (such as Turgeltaub's ALE or Nielsen's Brede methods). Very nice to see the learning curves in Fig 3! However, they lead me to the opposite conclusion, namely that the benefits of transfer learning are surprisingly small... Again, this may be related to the specific input chosen or the lack of modeling of output structure In conclusion, this is a solid piece of work- presenting real progress in the field